# Batch Fine Magnetic Pattern Transfer Method on Permanent Magnets Using Coercivity Change during Heating for Magnetic MEMS

**DOI:** 10.3390/mi15020248

**Published:** 2024-02-07

**Authors:** Keita Nagai, Naohiro Sugita, Tadahiko Shinshi

**Affiliations:** 1Department of Mechanical Engineering, Tokyo Institute of Technology, 4259 Nagatsuta-cho, Midori-ku, Yokohama 226-8503, Japan; nagai.k.an@m.titech.ac.jp; 2Laboratory for Future Interdisciplinary Research of Science and Technology (FIRST), Institute of Innovative Research (IIR), Tokyo Institute of Technology, 4259 Nagatsuta-cho, Midori-ku, Yokohama 226-8503, Japan; sugita.n.aa@m.titech.ac.jp

**Keywords:** laser-assisted heating, magnetic MEMS, magnetic pattern transfer, micromagnetization, multi-pole magnetization, NdFeB magnet

## Abstract

In magnetic microelectromechanical systems (MEMSs), permanent magnets in the form of a thick film or thin plate are used for structural and manufacturing purposes. However, the geometric shape induces a strong self-demagnetization field during thickness–direction magnetization, limiting the surface magnetic flux density and output power. The magnets must be segmented or magnetized in a fine and multi-pole manner to weaken the self-demagnetization field. Few studies have been performed on fine multi-pole magnetization techniques that can generate a higher surface magnetic flux density than segmented magnets and are suitable for mass production. This paper proposes a batch fine multi-pole magnetic pattern transfer (MPT) method for the magnets of MEMS devices. The proposed method uses two master magnets with identical magnetic patterns to sandwich a target magnet. Subsequently, the coercivity of the target magnet is reduced via heating, and the master magnet’s magnetic pattern is transferred to the target magnet. Stripe, checkerboard, and concentric circle patterns with a pole pitch of 0.3 mm are magnetized on the NdFeB master magnets N38EH with high intrinsic coercivity via laser-assisted heating magnetization. The MPT yields the highest surface magnetic flux density at 160 °C, reaching 39.7–66.1% of the ideal magnetization pattern on the NdFeB target magnet N35.

## 1. Introduction

Magnetic microelectromechanical systems (MEMS) devices utilizing electromagnetic forces, in contrast to electrostatic or piezoelectric MEMS devices, have large displacements and can be driven by low voltage. They do not require a voltage booster circuit and can be operated directly by a battery. Therefore, they have been applied in various fields, such as telecommunications [1], the automotive industry [2,3,4,5,6], medicine [7,8,9,10], and biotechnology [11].

However, the performance of magnetic MEMS devices lags behind that of devices based on alternative driving principles at scales of ≤1 mm. The high-performance magnets developed thus far are inadequate for enhancing the magnetic MEMS performance at this scale. This limitation arises from the uniform magnetization of thin-film or thick-plate magnets, which are characterized by low aspect ratios in the thickness direction, resulting in a strong self-demagnetization field within the magnet and a correspondingly low surface magnetic flux density. High-performance magnetic MEMS devices require high-performance magnets [12,13,14,15,16] and microfabrication and magnetization technologies. This is because the aspect ratio must be increased with the same thickness and width to match that in the magnetization direction while simultaneously weakening the self-demagnetization field in the magnet.

Magnets have been processed via dicing [17], ion milling [18], and wet etching [19] to increase their aspect ratios. Methods for fabricating high-aspect ratio and segmented magnets include using bonded magnets [20,21] and combining the liftoff method with magnet deposition, such as electrolytic plating [22], codeposition [13], and atomic layer deposition [12]. Even if the aspect ratio of the magnet is increased by segmentation, the self-demagnetization field can only be reduced if sufficient space is provided between the segments. This space limits the power density of the device.

One method for strengthening the magnetic field without using space or generating a self-demagnetization field, even when permanent magnets are placed close to each other, is multi-pole alternating magnetization. A laser-assisted heating (LAH) magnetization method was developed for the fine multi-pole magnetization of permanent magnets [23,24]. In this method, the localized area of a magnet in one direction is heated by a laser beam to reduce the coercivity, and the magnetization is reversed along an external magnetic field. The desired alternating magnetic pattern can be formed by repeating this method. However, the laser trajectory increases with the number of magnetized poles, the complexity of the pattern, and the magnetization time, making the method unsuitable for mass production.

The ultrahigh magnetization (UHM) process developed by Komura [25,26] is a fine multi-pole batch magnetization method for the mass production of micromotors. In this process, NdFeB magnets are heated to reduce their coercivity, and fine alternating magnetization is achieved by cooling them in a multi-pole magnetic field formed by an array of small SmCo magnets. However, the challenges in the microfabrication and assembly of SmCo magnets limit the realization of complex magnetic patterns. Furthermore, the NdFeB magnets must be heated to a Curie temperature of ≥330 °C, which exceeds the heat-resistance temperature of polymer materials used in MEMS devices. Therefore, the heating temperature must be kept below 200 °C for magnetic MEMS applications.

Thus, this study aimed to develop a magnetic pattern transfer (MPT) method that can transfer fine and complex magnetic patterns in batches below 200 °C to achieve enhanced performance in magnetic MEMS. This method overcomes the limitations of magnetic patterns by using fine and complex magnetization patterns fabricated using the LAH magnetization method as a substitute for SmCo magnet arrays in the UHM process.

Section 2 describes in detail the transfer principles and procedures of the MPT method. Section 3 describes the experimental methods, conditions, magnetic patterns, and sample magnets. As discussed in Section 4, the versatility of the MPT method was demonstrated by fabricating and transferring transfer–source magnetic patterns of stripes, checkerboards, and concentric circles. Section 5 presents the conclusions.

## 2. MPT Method

The principle of the proposed MPT method is illustrated in Figure 1. This method uses two master magnets that hold magnetic patterns during heating. The pattern of the master magnet is transferred to the target magnet. The pattern of the master magnet is formed via LAH magnetization [25], which allows for the formation of various micromagnetic patterns.

First, a target magnet—magnetized in the thickness direction—is sandwiched between two master magnets. When all the magnets are heated, the coercivity of the target magnet is lower than that of the master magnet because the intrinsic coercivity of the target magnet is lower than that of the master magnet. The magnetic pattern of the master magnets is then transferred to the target magnet. At this time, there are regions of the target magnet where the magnetization directions coincide with that of the master magnet and regions where the magnetization directions are opposite. In the regions where the magnetization directions coincide, the magnetic field generated by the master magnet increases the permeance coefficient of the target magnet, and the magnetic flux density at the operating point exceeds the flux density at the knick point, suppressing thermal demagnetization. In the region where the magnetization direction is the opposite, the magnetization is reversed by the leakage flux generated from the magnetic poles adjacent to this region, in addition to the magnetic field of the master magnet. The sum of the magnetic field of the master magnet and the leakage flux generated from the magnetic poles adjacent to this region is defined as the transfer magnetic field. Therefore, the magnetic pattern can be transferred at a lower temperature than that in the UHM process using the leakage flux of the target magnet.

## 3. Experimental Methods

### 3.1. Master Magnet Design

#### 3.1.1. Selection of the Master Magnet Material and LAH Magnetization Conditions

First, the materials used for the master magnet were investigated. The master magnet must be able to generate a strong magnetic field with little demagnetization, even when heated, and must have little irreversible demagnetization due to repeated changes in temperature. Therefore, the SmCo magnet sample with the highest heat resistance due to the high Curie temperature at which ferromagnetism disappears was selected as a candidate for the master magnet material. In addition, a magnet sample with high heat resistance and so-called high intrinsic coercivity among NdFeB magnets, known to have the strongest magnetic force, was also selected. These permanent magnets were subjected to fine multi-pole magnetization using the LAH magnetization method, and a magnet with a high surface magnetic flux density was selected as the master magnet material.

Table 1 presents the catalog values of the magnetic properties of the candidate master magnet materials. The magnet samples were machined via wire electrical discharge machining (EDM) to 5 mm × 5 mm × t0.3 mm using a sintered bulk magnet. Under the LAH magnetization method conditions presented in Table 2, the laser scanning speed at which the surface magnetic flux density was maximized was determined for each master magnet sample. As shown in Figure 2, the surface of the magnetic sample was divided into eight regions. The laser scanning speed was varied in each region, the speeds differed among the regions, and seven poles were alternately magnetized with a magnetization width of 0.3 mm. The optimal sequence of laser irradiation depends on the magnetic pattern and magnet geometry; however, in this experiment, the magnets were irradiated from the left end. With reference to Figure 2, the first row on the left side of regions (1), (3), (5), and (7) was heated in a straight line at different rates. Laser scanning was performed eight times for each region. After the magnets had cooled completely, the next row on the right side, which was 0.6 mm away, was heated. The distance between the fourth and fifth irradiation lines from the odd to even regions was 1.2 mm. This process was repeated until four lines were irradiated to obtain four irradiation lines in each region. As shown in Figure 3, several sintered bulk NdFeB magnets (N35, MagFine Corp., Saitama, Japan) were combined to generate an external magnetic field along the thickness of the magnetic samples. The external magnetic field was set to −0.7 T for the SmCo magnet and −0.9 T for the NdFeB magnet in the out-of-plane downward direction. The magnet sample was then fixed with Kapton tape to a magnet that generated an external magnetic field.

The surface magnetic flux density on the centerline of the divided region was measured for the multi-pole magnetized magnet samples using the measurement system shown in Figure 4. In this system, a Hall probe (HG-0711, Asahi Kasei Microdevices Corp., Tokyo, Japan) with an out-of-plane directional sensitive area of 50 µm × 50 µm was placed in contact with the magnet sample surface fixed on a linear stage, and the stage was moved by a piezo motor (M-227.25, Physik Instrumente GmbH & Co. KG, Karlsruhe, Germany) to perform the measurements.

Figure 5 shows the measurement results for the surface magnetic flux density in the region magnetized at laser scanning speeds of 150 and 160 mm/s for the NdFeB magnet, which allowed for the measurement of two regions on a single measurement line, as shown in Figure 2. The laser scanning speed that resulted in the maximum surface flux density was applied to magnetize the master magnet. The peak-to-peak (p-p) values of the pole pairs with a maximum surface flux density were measured near the center of each region with small edge effects in the ranges of 0.9–1.5 mm and 3.5–4.1 mm.

#### 3.1.2. Master Magnet Fabrication

The magnet materials and magnetizing conditions described in Section 3.1.1 were used to fabricate master magnets with three different magnetic patterns, as shown in Figure 6: stripe, checkerboard, and concentric circles. The laser scan trajectories are shown in Figure 7. The laser scan trajectories in the stripe pattern were heated in the same sequence, as described in Section 3.1. For the checkerboard and concentric circle patterns, the magnetization conditions and widths were equal to those of the stripe pattern. The arc length of the concentric circles was equal to the length of the striped pattern. The laser scan length at one pole of the checkerboard was equal to the magnetization width, and shorter than that of the striped pattern.

The magnetization results for the master magnet were obtained using the measured and simulated surface magnetic flux densities over the entire surface. As shown in Figure 8, a Hall element was scanned at 0.3 mm intervals from a position 0.1 mm from the magnet edge, and the surface magnetic flux density on each straight line was measured. The surface magnetic flux density was simulated via three-dimensional (3D) magnetic field analysis (Ansys Electronics Desktop 2021 R1, Ansys Inc., Canonsburg, PA, USA), modeled with the same shape and target pattern as the master magnet, as shown in Figure 6. The properties of the master magnets used in the simulations are presented in Table 1. The root mean square (RMS) values of the measured and simulated surface flux densities over the all surface of magnet were calculated. The ratio of the RMS of the measured value to that of the simulated value was defined as the magnetization ratio of the magnetic pattern. The average of the measured surface flux density over all the magnet surfaces was defined as the offset. These calculations were used to evaluate the magnetization results of the master magnet.

### 3.2. MPT TEST

First, the transfer temperature in the MPT test was surveyed. A NdFeB magnet with dimensions of 10 mm ×10 mm × t0.5 mm was used as the target magnet. Table 3 presents the catalog values of the magnetic properties of the target magnet. The transfer temperature was defined as the temperature at which the surface magnetic flux density of the transferred target magnet reached its maximum value. The magnetization pattern of the master magnet was striped. The surface magnetic flux density of the target magnet was measured along a straight line located 5.625 mm from the edge of the target magnet, which was the center of the line where regions (5) and (6) of the master magnet were transferred to avoid the influence of the edges. The determined transfer temperatures were applied to the MPT tests of the other patterns.

Figure 9 shows the MPT test setup and process. The experiment was performed at room temperature (22 °C) and atmospheric pressure. As shown in Figure 9a, a non-magnetic steel block (JIS SUS303) was placed on a hot plate for temperature stabilization, and the target magnet sandwiched between two master magnets was placed on the steel block. The magnets on the steel block were then covered with an aluminum container and heated.

In the heating process shown in Figure 9b, first (1), the surface temperature of the steel block was measured using a radiation thermometer, and then the steel block was heated to the target temperature of the steel block. (2) When the surface temperature stabilized, the magnets were placed at the center of the steel block and covered with an aluminum container. (3) After the magnets were heated for a specified duration, the magnetic sample was removed from the block and cooled to room temperature. The process involved an experimental temperature that varied from 100 to 200 °C, as well as the surface temperature of the steel block before and after being covered with the aluminum container. The experimental temperature and surface temperature of the steel block covered with an aluminum container had to match. However, because the thermocouples could not be placed at the center of the steel block where the magnets were placed, the temperature after covering the steel block was estimated from the surface temperature of the steel block before it was covered with the aluminum container.

To estimate the surface temperature of the steel block after it was covered with an aluminum container, the emissivity and relationship between the surface temperatures of the steel block before and after it was covered with the aluminum container were examined. The emissivity ε was calibrated as 0.2 from the thermocouple measurements on the surface of the steel block, as shown in Figure 9c. After the steel block was covered with an aluminum container for 200 s, the surface temperature of the block stabilized. The heating time was set to 300 s with a margin.

The transfer results of the target magnet were evaluated using the measured and simulated surface magnetic flux densities over all the magnet surfaces. The simulation was completely magnetized according to the target pattern shown in Figure 6, and the magnetic properties of the target magnet listed in Table 3 were used. As shown in Figure 10, the Hall element was scanned at 0.3 mm intervals from 2.4 mm from the magnet center to measure the surface magnetic flux density on 17 straight lines. To simulate the surface magnetic flux density of the target magnet, the model mentioned in Section 3.1.2 was used; only the magnetic properties were changed, and the surface magnetic flux density at the same location was calculated. As described in Section 3.1.2, the RMS values of the measured and simulated surface flux densities over the entire surface of the target magnet were calculated. The ratio of the RMS of the measured value to that of the simulated value was defined as the transfer ratio of the magnetic pattern used to evaluate the transfer results of the target magnet. Figure 4 shows the surface magnetic flux density measurement system. The magnetic flux density above 100 µm of the magnet surface was measured by tracing the magnet using a Hall element (Asahi Kasei Electronics, HG-0711, Tokyo, Japan) with a sensitive area of 50 µm × 50 µm at the tip of the probe.

For each magnet sample, the ratio of the measured and simulated RMS values of the surface magnetic flux density was calculated over all the magnet surfaces, as was the magnetization ratio, defined as the transfer ratio. The offset of the magnetic pattern was also calculated in the same way as for the master magnet in Section 3.1.2, and these were used to evaluate the transfer results.

## 4. Experimental Results and Discussion

### 4.1. Experimental Master Magnet and LAH Magnetization Conditions

Figure 11 shows the relationship between the surface magnetic flux density p-p and the laser scanning speed for the master magnet material candidates. The NdFeB magnet N38EH (NeoMag Corp., Tokyo, Japan) had a higher surface flux density than the SmCo magnet SS30H (NeoMag Corp., Tokyo, Japan). As shown in Figure 11, the surface magnetic flux density of the SmCo magnet increased as the scanning speed decreased. These results suggest that the surface magnetic flux density would increase if the laser scanning speed was reduced below 100 mm/s. Therefore, the laser scanning speed was reduced, but the magnet sample broke owing to the stress concentration caused by local thermal expansion.

A thermal stress analysis (Ansys Workbench 2021 R1, Ansys Inc.) of the SmCo magnet was performed using the simulation model and conditions shown in Figure 12 and Table 4. The heat transfer analysis with the laser movement was simplified. Instead of continuously moving the heating region, the simulation was conducted by dividing the heating region and switching it at a time that suited the scan position and speed. The model’s width was set to 3 mm as a sufficient length for the analysis, compared to a target magnetization width of 0.3 mm. Because the laser scan length and spot diameter in the experiment were 1.3 and 0.1 mm, respectively, the heating dimensions of one area in the model were 0.163 mm in the scan direction and 0.1 mm in the scan width, respectively. The laser scan length was divided into eight regions.

Figure 13 shows the simulated relationships among the laser scanning speed, heating temperature, and thermal stress in region (4), as shown in Figure 12. The average temperature in Figure 13a is the temperature in the SmCo magnet and is the average value of the line in the thickness direction passing through the center of region (4), as shown in Figure 12. In Figure 13b, for each laser scanning speed, it is the maximum value of the thermal stress in a line on the magnet surface perpendicular to the laser scanning direction and passing through the center of region (4). Figure 13a shows that the temperature of the SmCo magnet increases as the laser scanning speed decreases. So, a slower speed than the 100 mm/s range explored in this experiment will result in higher heating temperatures, and magnetization reversal can easily occur. However, Figure 13b indicates that the thermal stress exceeds the SmCo magnet breaking stress of 40 MPa and the SmCo magnet breaks when the laser scanning speed is <50 mm/s. The SmCo magnets broke at a laser scanning speed of <100 mm/s in the experiment. This difference was due to the residual stress generated during the wire EDM. Therefore, increasing the surface magnetic flux density of SmCo magnets is difficult because a reduction in the laser scanning speed destroys the magnet. Therefore, a master magnet was fabricated using the NdFeB magnet N38EH.

As shown in Figure 14, an appropriate laser scanning speed of 160 mm/s was selected to maximize the p-p of the surface flux density, and the offset was close to zero. An offset close to zero indicates good alternating magnetization, with equal positive and negative magnetization.

### 4.2. Master Magnet Fabrication Results

Figure 15 shows the measured and simulated surface magnetic flux densities for each patterned master magnet. Table 5 presents the magnetization ratios and offsets. As shown in Figure 15, the measured patterns were close to the simulated patterns. The magnetization ratio of each pattern was as high as 70–80%. However, as shown in Table 5, the offset of the stripe and checkerboard pattern showed a more significant different. The large offset is due to overheating caused by the overlap of the heating area with the spot diameter at the position where the laser scanning is divided in the laser scanning trajectory, as shown in Figure 7. The offset of the concentric circles pattern has a positive value depending on the four corner regions.

The magnetization conditions of each pattern, such as the scanning speed, magnitude of the applied magnetic field, and laser scan path, should be investigated to improve the magnetization ratios and achieve a measured offset close to zero.

### 4.3. Magnetic Pattern Transfer Results

The experimental results of the stripe pattern transfer are shown in Figure 16 and Figure 17. Figure 16 shows the experimental surface magnetic flux density distribution of the striped pattern on the target magnet at each temperature. Figure 17 shows that for the stripe pattern transfer, under heating at 160 °C, the p-p value of the surface flux density was maximized, and the offset value was relatively close to zero. Therefore, the transfer temperature for this test was determined to be 160 °C.

Figure 18 shows the measured and simulated surface magnetic flux densities for magnets with three different transferred magnetic patterns. The simulation results are based on a fully magnetized model as in the target magnetic pattern, as described in Section 3.2, and do not reproduce the transfer experiment with the modeling of the fabricated master magnet. The measured patterns are similar to the simulated patterns. Table 6 presents the transfer ratios and offsets for each pattern. In Figure 18, the transfer ratio of each pattern is 39.7–66.1%. Compared to the master magnet, the offset improved for the stripe pattern to −7.9 mT and degraded for the concentric circles pattern to −6.4 mT.

### 4.4. Discussion of the Magnetic Pattern Transfer

The reasons why the measured offset of the target magnet was not equal to the simulated offset and the low transfer ratio are discussed below. First, an offset of the target magnet exists depending on the offset of the master magnet. For the stripe and checkerboard patterns, the offset of the target magnet was improved over the offset of the master magnet, but the offset was still significant. As shown in Figure 19, the improvement in the offset is due to the transfer magnetic field in the region where the poles of the master magnet are switched, as they are weaker than the magnetic field H_Rev_ required for the magnetization reversal to occur. However, the region where the transfer magnetic field is weak is demagnetized; so, the offset of the target magnet remains. In addition, in the regions at the four corners of the concentric circle pattern, the transfer magnetic field of the master magnet is too weak to transfer because of the low aspect ratio. Therefore, the offset of the concentric circle pattern is negative.

Next, the low transfer ratio is possibly the weak transfer magnetic field. The magnetic field transfer during the heating process, as shown in Figure 1b step 2, was calculated using the simulation model shown in Figure 20 through a 3D magnetic field analysis. In this simulation, the magnetic properties of the master magnet in the magnetization reversal region were initially adjusted to approach the surface magnetic flux density distribution measured at room temperature. In the magnetization ratio calculations, the catalog values were used for the magnetic properties of the master magnet. However, in this simulation, the magnetic properties of the master magnet in the magnetization reversal region were identified as B_r_ = 0.44 T and H_c_ = 356.8 kA/m to match the measured surface magnetic flux density. At the same time, catalog values were used for other regions.

The magnetic properties of the master magnet at different temperatures were calculated using the identified B_r_ and H_c_ values, corrected using the temperature coefficients presented in Table 1. The magnetic properties of the target magnet at different temperatures were calculated using the catalog values of B_r_ and H_c_ at room temperature and the temperature coefficients presented in Table 2.

The calculated transfer magnetic field H_trans_ was the average value on a line (lines A–A in Figure 20) in the thickness direction passing through the center of the magnetization-reversal region of the target magnet. The intrinsic coercivity Hcj of the target magnet and the magnetic field H_sat_ at which its magnetization was saturated were obtained from the J–H curve measured using a pulsed high-field flux meter (TPM, TPM-2-08s25VT, Toei Industry Co., Ltd., Tokyo, Japan), as shown in Figure 21. The easy axis of the magnetization of the sample was aligned with the direction of the applied magnetic field.

Figure 22 shows the relationships between the heating temperature and the simulated transfer magnetic field H_trans_, the measured intrinsic coercivity H_cj_, and the measured saturation magnetic field H_sat_. At a heating temperature of 120 °C, the transfer magnetic field H_trans_ surpassed the intrinsic coercivity H_cj_, initiating the magnetization reversal in the region shown in Figure 20. The results in Figure 16 confirm that the positive surface magnetic flux density increased from 120 °C in the region where magnetization reversal was desired and where magnetization reversal occurred. As shown in Figure 17, the offset increases after 160 °C, the amplitude of the magnetization reversal region decreased, and the effect of the thermal demagnetization became more pronounced than that of the magnetization reversal. Thus, the effect of heating in the MPT method is, first, that as temperature increases, magnetization reversal is more likely to occur, and conversely, demagnetization increases. Due to the balance between these two phenomena, there is a specific temperature at which the surface flux density is at a maximum, and the offset approaches zero. At this time, the offset is not necessarily zero because the temperature at which magnetization reversal occurs sufficiently and the temperature at which the effects of thermal demagnetization become pronounced vary according to the heat resistance of the target magnet. The results in Figure 17 suggest that the temperature at which thermal demagnetization becomes pronounced is lower than the temperature at which sufficient magnetization reversal occurs.

For complete magnetization reversal, it is essential that H_trans_ exceeds or equals H_sat_. However, Figure 22 suggests that H_trans_ was smaller than H_sat_ in this experiment. Thus, for improving the transfer ratio below 200 °C, the options include increasing H_trans_ or reducing H_sat_. Increasing H_trans_ involves switching to NdFeB magnets with a high residual magnetic flux density for both master and target magnets. Reducing H_sat_ involves using a target magnet with a low H_cj_ at room temperature. However, magnets with small H_cj_ values are generally susceptible to irreversible thermal demagnetization. Therefore, reducing H_sat_ may not contribute to an improvement in the transfer ratio. For thermal demagnetization, it is necessary to further simulate the cooling process in step 3 of Figure 1b. However, simulation is challenging because magnetization reversal and thermal demagnetization occur simultaneously.

## 5. Conclusions

This paper proposes a batch MPT method for fine multi-pole magnetic patterns in magnetic MEMS devices. In this process, the target magnet, which is sandwiched between two master magnets with the original magnetic patterns, is heated. The coercivity of the target magnet is then reduced, and the magnetic pattern of the master magnet is transferred in a batch.

The master magnet must generate a strong magnetic field during the heating process and have little irreversible demagnetization even after repeated heating and cooling. Therefore, the SmCo magnet SS30H with a high Curie temperature and the NdFeB magnet N38EH with a high intrinsic coercivity Hcj were selected as the master magnet candidates. These magnetic materials were magnetized in a stripe pattern (magnetization width of 0.3 mm) using the LAH magnetization method while varying the laser scanning speed from 100 to 200 mm/s as the magnetization condition. Consequently, NdFeB magnets with high surface flux densities were selected as the master magnet materials.

SmCo magnets tended to have higher surface flux densities at lower scanning speeds but broke down at speeds of <100 mm/s. Thermal stress analysis indicated that the thermal stress of the SmCo magnet exceeded the breaking stress at a scanning speed lower than 50 mm/s, although errors were introduced because the residual stress during machining was not considered. Therefore, SmCo was not selected as the master magnet material.

Master magnets of the NdFeB magnet with stripe, checkerboard, and concentric circular patterns with a magnetization width of 0.3 mm were fabricated. These master magnets were then used to apply the MPT method at a heating temperature of 160 °C. The transferred magnetic patterns agreed closely with the simulation, and a 39.7–66.1% transfer ratio was achieved.

To increase the transfer ratio, it is necessary to either strengthen the transfer magnetic field or weaken the magnetic field at which the magnetization of the target magnet is saturated. To weaken the magnetic field at which the target magnet is saturated, it is necessary to reduce the intrinsic coercivity of the target magnet. However, this approach is not promising, because it can easily lead to irreversible thermal demagnetization of the target magnet. In the future, to improve the transfer ratio, the transfer magnetic field should be enhanced using a master magnet and a target magnet with a higher residual magnetic flux density.

## Figures and Tables

**Figure 1 micromachines-15-00248-f001:**
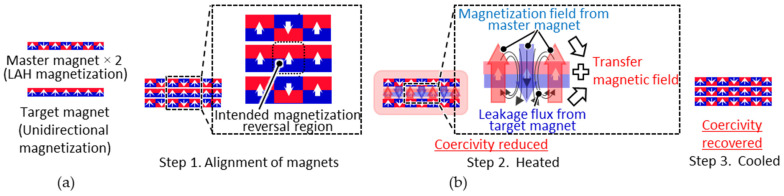
Magnetic pattern transfer (MPT) method: (**a**) two master magnets with magnetic patterns and a target magnet unidirectionally magnetized in the thickness direction; (**b**) transfer procedure: Step 1: the target magnet is sandwiched between the two master magnets; step 2: these magnets are heated to reduce the coercivity of the target magnet. The magnetization of the intended region in the target magnet is then reversed by a transfer magnetic field consisting of the magnetic field of the master magnet and the leakage flux of the adjacent magnetic poles in the region where the magnetization reversal of the target magnet is intended; step 3: these magnets are cooled naturally, the coercivity of the target magnet is recovered, and the magnetic pattern transfer is achieved.

**Figure 2 micromachines-15-00248-f002:**
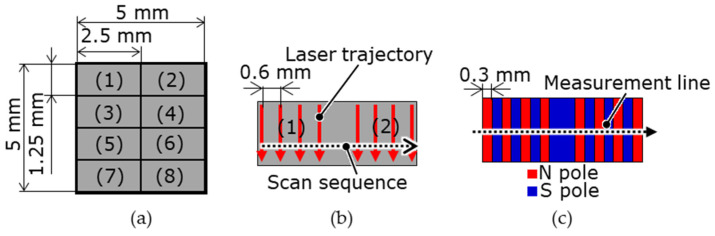
Segmentation of the magnet surface and laser scan trajectories for the magnetization condition search: (**a**) eight divided areas of the magnet; (**b**) the laser scan trajectories have four paths with a pitch of 0.6 mm in each segmented area, starting from the left end; (**c**) the magnetization pattern is eight poles in each area.

**Figure 3 micromachines-15-00248-f003:**
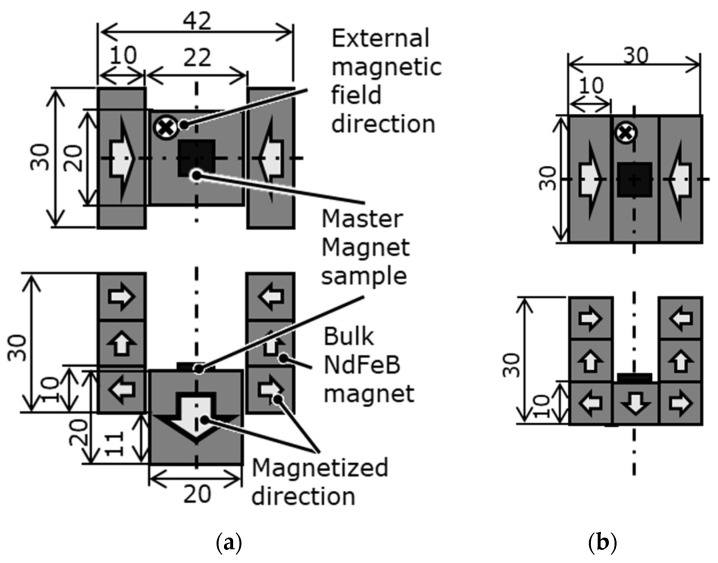
Uniform external magnetic field generated by multiple bulk NdFeB magnets in the out-of-plane direction to a master magnet sample: (**a**) 0.7 T; (**b**) 0.9 T.

**Figure 4 micromachines-15-00248-f004:**
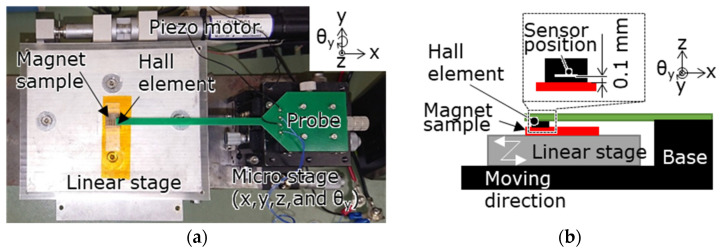
Surface magnetic flux density measurement system consisting of a piezo-motor-driven linear stage scanning magnetized area over a magnet with a Hall probe and a 4-DOF micro stage to adjust the probe position and attitude: (**a**) photograph; (**b**) schematic.

**Figure 5 micromachines-15-00248-f005:**
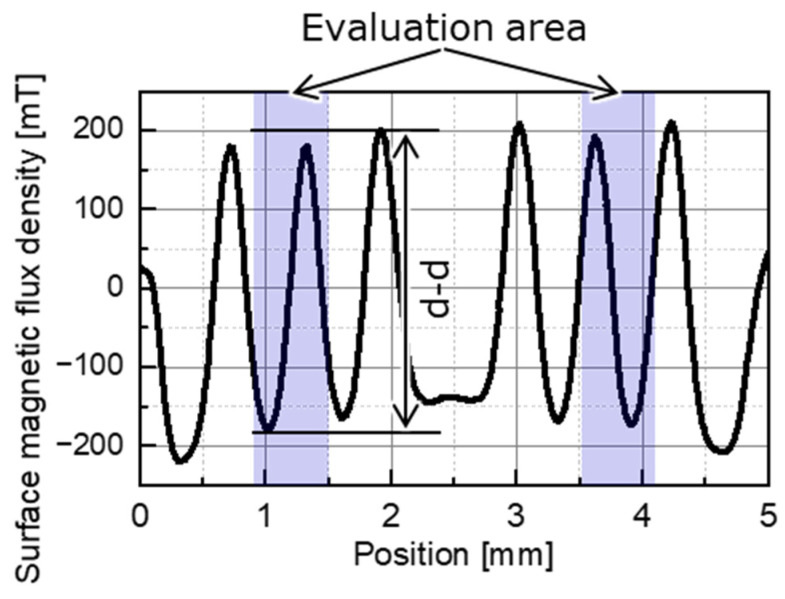
Measured surface flux density distribution of an area magnetized on an N38EH magnet at a scanning speed of 150–160 mm/s.

**Figure 6 micromachines-15-00248-f006:**
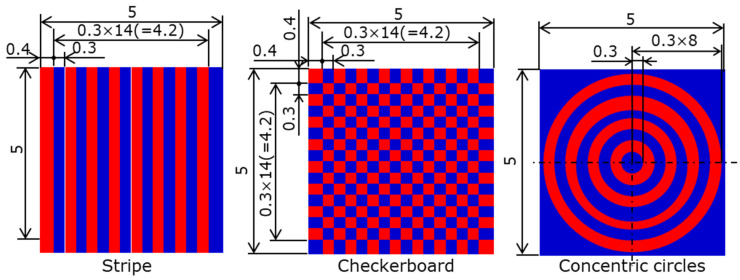
Magnetization patterns of the master magnets.

**Figure 7 micromachines-15-00248-f007:**
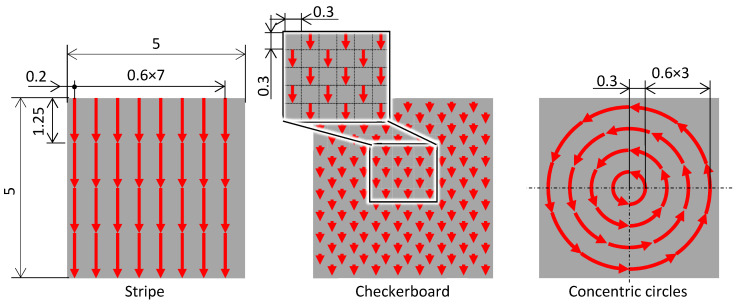
Laser scan trajectories of each magnetization pattern for the master magnets.

**Figure 8 micromachines-15-00248-f008:**
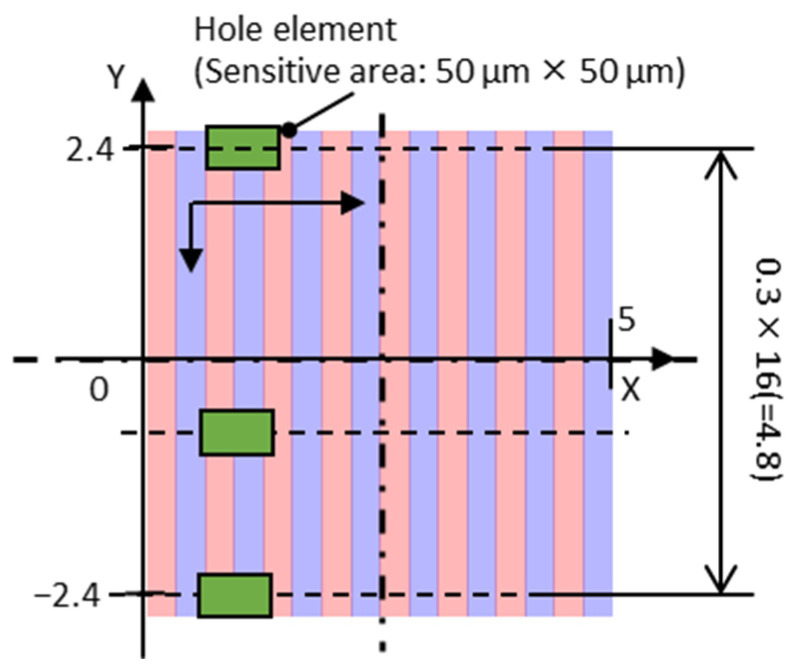
Magnetic flux density measurement on the master magnet surface, continuous horizontal scanning of the Hall probe, and vertical positioning at a pitch of 0.3 mm.

**Figure 9 micromachines-15-00248-f009:**
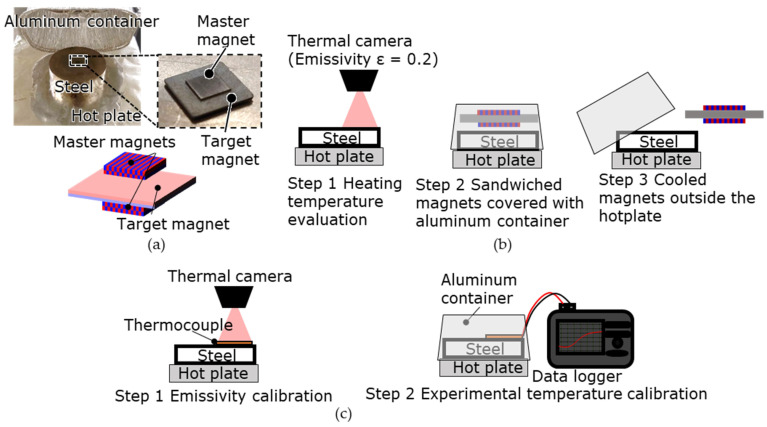
Experimental setup and process of the MPT test: (**a**) experimental photo and configuration consisting of a hot plate, a steel block, and an aluminum container. (**b**) Experimental process: step 1: the surface temperature on the steel block is measured with a thermal camera; step 2: a target magnet sandwiched by two master magnets on the steel block is heated with an aluminum container; step 3: these magnets are removed from the hot plate and naturally cooled. (**c**) Calibration of emissivity and temperature: step 1: calibration of the emissivity of the thermal camera using a thermocouple placed on the surface of the steel block; step 2: determination of heating time by measuring the time required for the temperature inside the container to stabilize.

**Figure 10 micromachines-15-00248-f010:**
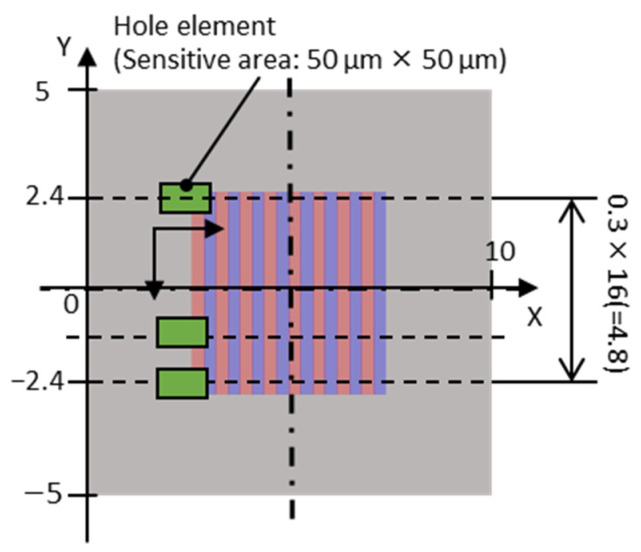
Magnetic flux density measurement on the target magnet surface, continuous horizontal scanning of the Hall probe, and vertical positioning at a pitch of 0.3 mm.

**Figure 11 micromachines-15-00248-f011:**
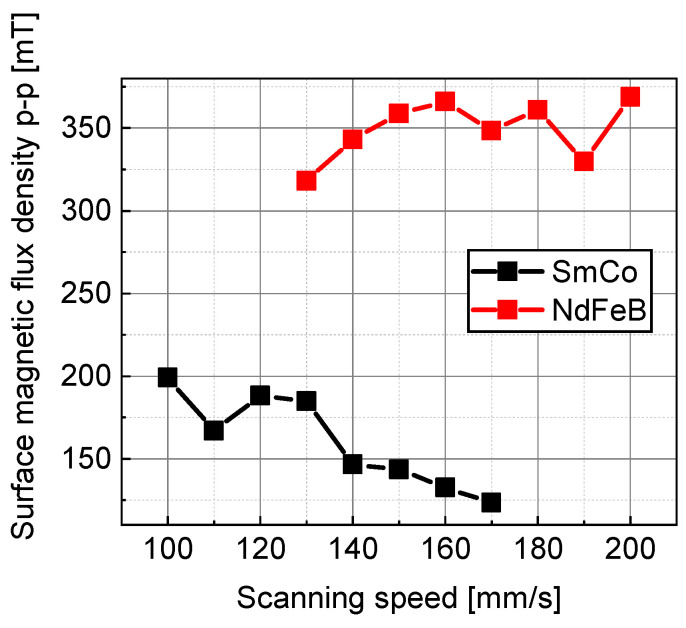
Experimental relationship between the scanning speed and the surface magnetic flux density p-p for the master magnet candidates.

**Figure 12 micromachines-15-00248-f012:**
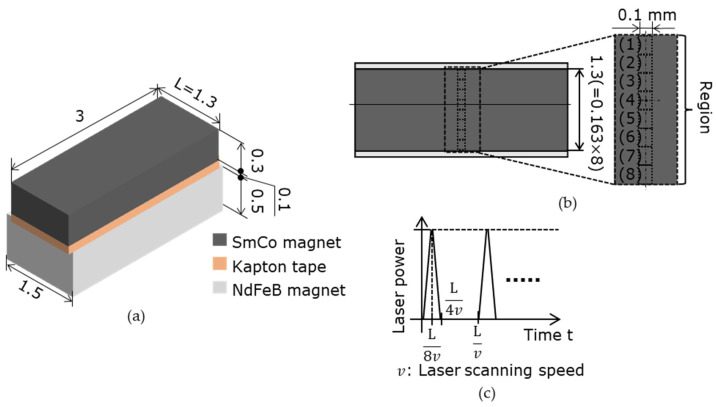
Thermal stress simulation model of the LAH method: using a simple calculation by changing the timing of the heating area following the scanning speed: (**a**) the simulation model consists of three layers: a SmCo magnet, Kapton tape to fix the sample, and NdFeB to generate an external magnetic field; (**b**) top view of the model, divided into eight areas at the center; (**c**) time chart of laser heating in the area (1).

**Figure 13 micromachines-15-00248-f013:**
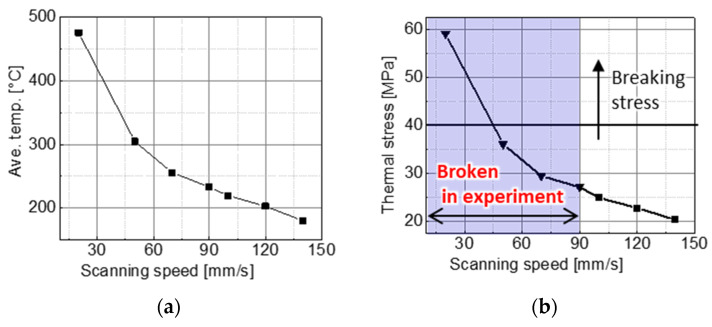
Average heating temperature and thermal stress with respect to the laser scanning speed: (**a**) relationship between the laser scanning speed and the average heating temperature; (**b**) relationship between the laser scanning speed and the thermal stress.

**Figure 14 micromachines-15-00248-f014:**
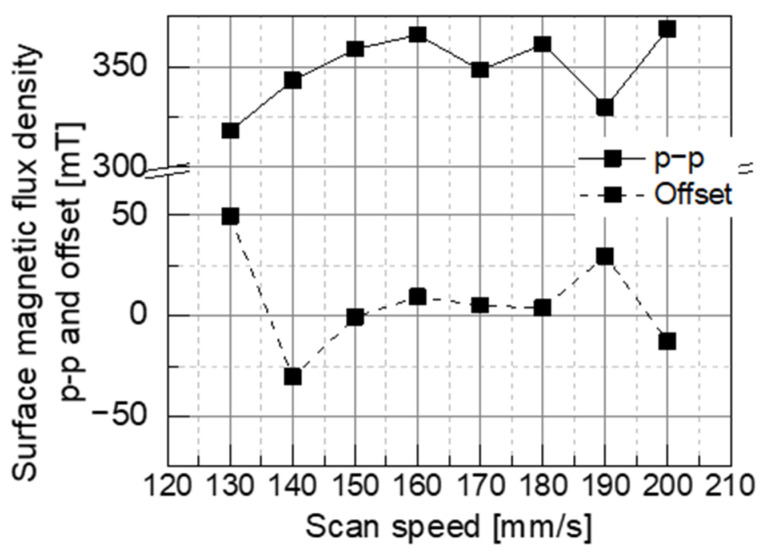
Surface magnetic flux density p-p and offset of the NdFeB magnet N38EH.

**Figure 15 micromachines-15-00248-f015:**
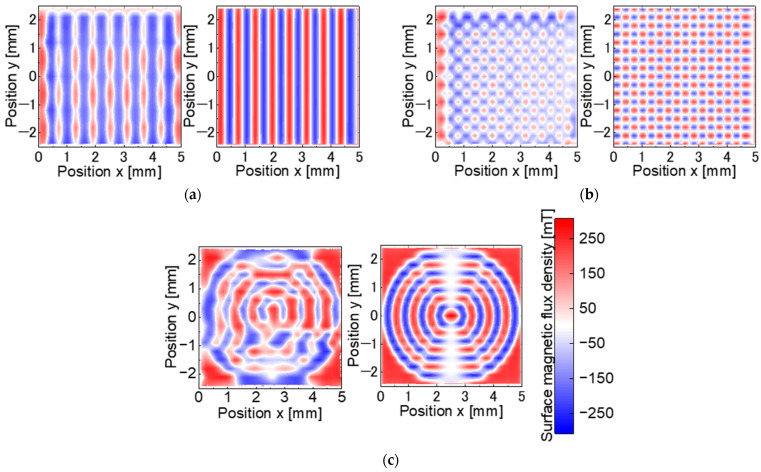
Surface magnetic flux density distribution of the master magnets: (**a**) stripe pattern; (**b**) checkerboard; (**c**) concentric circles.

**Figure 16 micromachines-15-00248-f016:**
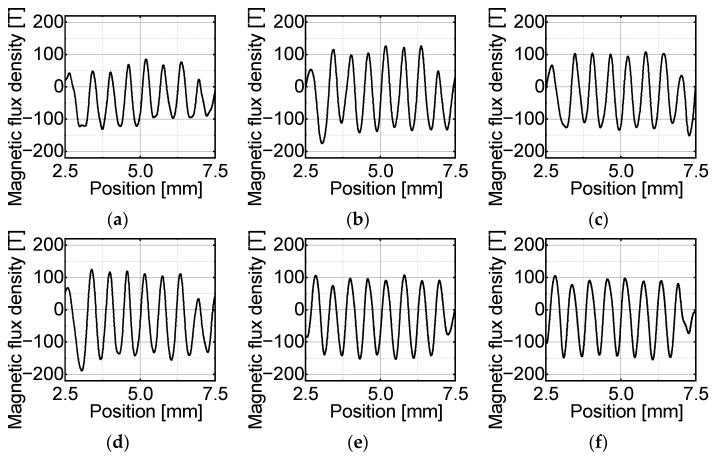
Surface magnetic flux density distribution of the stripe pattern at each experimental temperature: (**a**) 100 °C; (**b**) 120 °C; (**c**) 140 °C; (**d**) 160 °C; (**e**) 180 °C; (**f**) 200 °C.

**Figure 17 micromachines-15-00248-f017:**
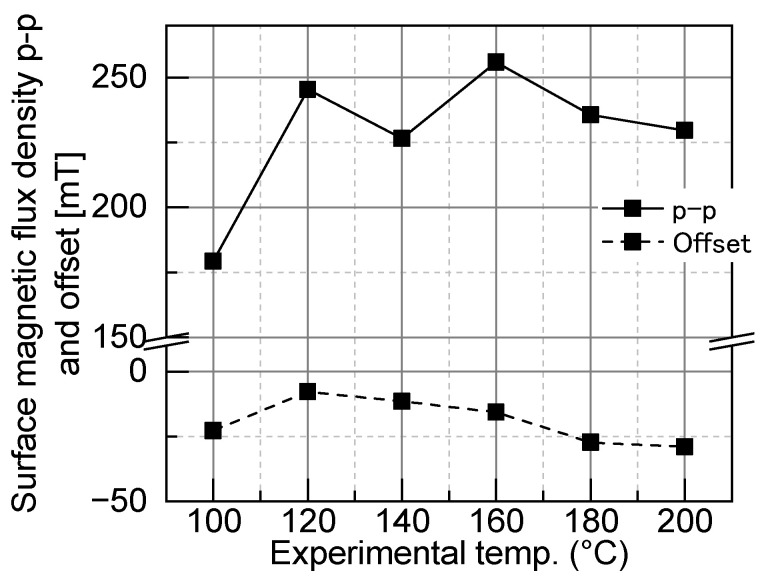
Surface magnetic flux density p-p for each experimental temperature.

**Figure 18 micromachines-15-00248-f018:**
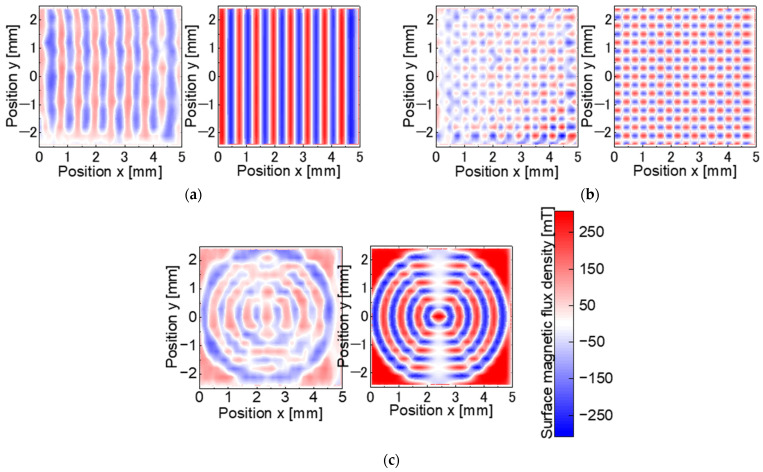
Surface magnetic flux density distribution of the target magnets: (**a**) stripe pattern; (**b**) checkerboard; (**c**) concentric circles.

**Figure 19 micromachines-15-00248-f019:**
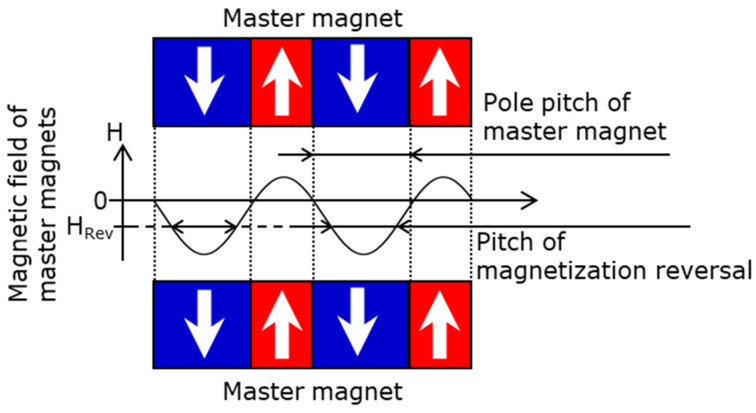
Relationship between the pole pitch and offset of the master magnet and its effect on the magnetization reversal pitch.

**Figure 20 micromachines-15-00248-f020:**
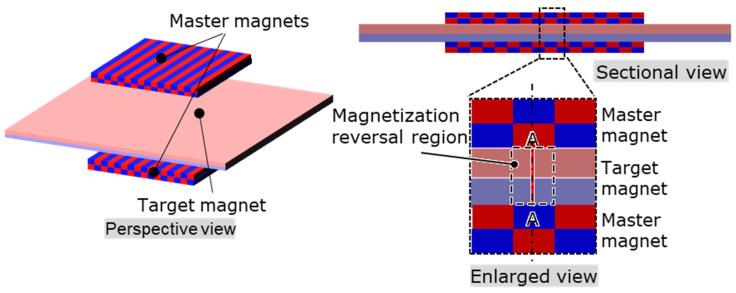
Simulation model for the transfer magnetic field calculation in the MPT method using a stripe pattern master magnet.

**Figure 21 micromachines-15-00248-f021:**
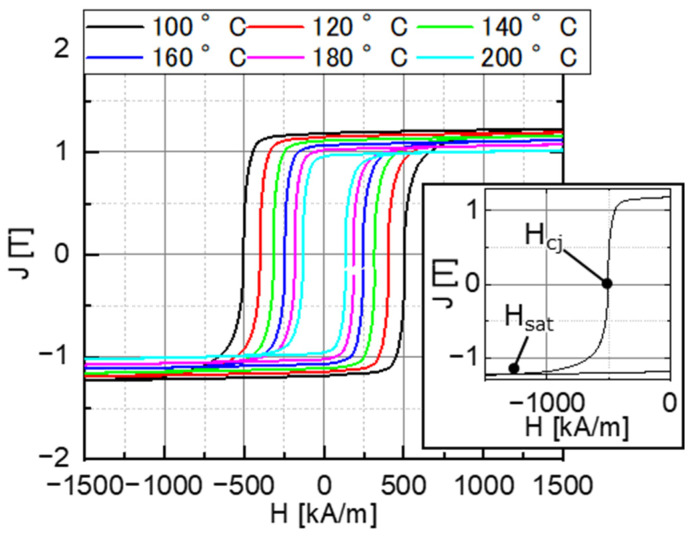
Measured magnetization curves of the target magnet using the TPM.

**Figure 22 micromachines-15-00248-f022:**
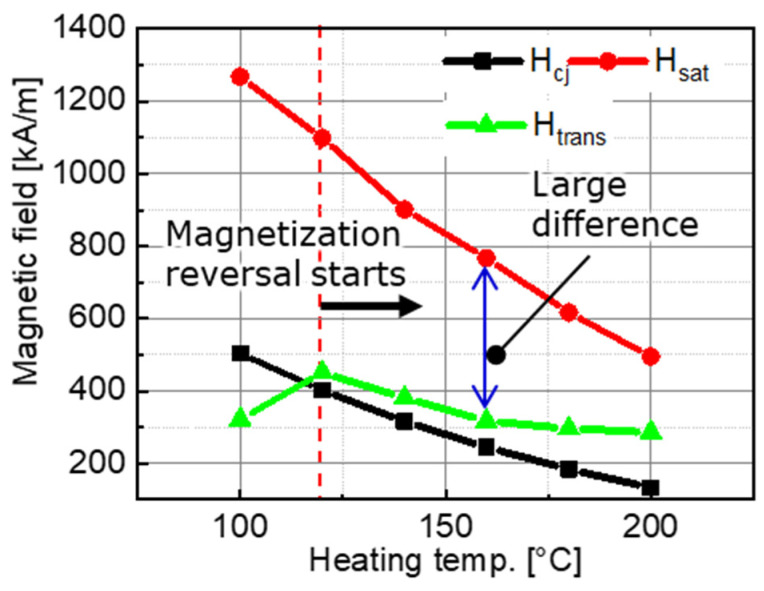
Relationships between the heating temperature and the measured intrinsic coercivity H_cj_, magnetic field for the magnetic saturation H_sat_, and the transfer magnetic field H_trans_.

**Table 1 micromachines-15-00248-t001:** Magnetic properties of the master magnets.

Magnet Type	Residual Flux Density B_r_ [T]	Coercivity H_c_ [kA/m]	Intrinsic Coercivity H_cj_ [kA/m]	Temperature Coefficient [%/°C]	
				B_r_	H_cj_
NdFeB N38EH	1.24	907	2388	−0.04	−0.2
SmCo SS30H	1.09	830	1990	−0.03	−0.15

**Table 2 micromachines-15-00248-t002:** LAH magnetization conditions for the magnetization condition search.

Laser Type	Repetition Frequency [kHz]	Spot Diameter [mm]	Scan Pitch [mm]	Laser Power [W]	Scanning Speed [mm/s]	External Mag. Field [T]
YVO_4_ 532 nm	30	0.1	0.6	6	100–200	0.7, 0.9

**Table 3 micromachines-15-00248-t003:** Magnetic properties of the target magnets.

Magnet Type	Residual Flux Density B_r_ [T]	Coercivity H_c_ [kA/m]	Intrinsic Coercivity H_cj_ [kA/m]	Temperature Coefficient [%/°C]	
				B_r_	H_cj_
NdFeB N35	1.20	870	955	−0.12	−0.55

**Table 4 micromachines-15-00248-t004:** Thermal stress analysis conditions for LAH method.

	Density [kg/m^3^]	Thermal Expansion Coefficient [kA/m]	Young’s Modulus [GPa]	Poisson’s Ratio	Thermal Conductivity [W/m K]	Specific Heat Capacity [J/kg K]
SmCo	8400	C⊥ 1.0 × 10^−5^ C// 8.0 × 10^−6^	151	0.3	23	360
Kapton tape	1420	2.7 × 10^−5^	3.4	0.3	0.16	1.1
NdFeB	7400	C⊥ −1.5 × 10^−6^ C// 6.5 × 10^−6^	166	0.3	8.9	500

**Table 5 micromachines-15-00248-t005:** Magnetization ratio and offset of master magnets.

	Stripe	Checkerboard	Concentric Circles
Magnetization ratio [%]	68.9	69.7	80.1
Offset [mT]	−50.2	−28.2	−28.2

**Table 6 micromachines-15-00248-t006:** Transfer ratio and offset of target magnets.

	Stripe	Checkerboard	Concentric Circles
Magnetization ratio [%]	45.3	66.1	39.7
Offset [mT]	−16.8	−7.3	−16.2

## Data Availability

Data are contained within the article.

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
