# Peer review of "Batch Fine Magnetic Pattern Transfer Method on Permanent Magnets Using Coercivity Change during Heating for Magnetic MEMS"

_micromachines, 2024, doi:10.3390/mi15020248_

Round 1

Reviewer 1 Report

Comments and Suggestions for Authors

This manuscript presents the study of a fast batch magnetization method termed multi-pole magnetic pattern transfer (MPT) for fine polarity pattern formation in a target magnet through master magnets served as molds of magnetic pattern. The contents are rich with experimental and simulation results. The novelty and importance of this manuscript is very high in the related community. However, the presentation and English writing style need much improvement before being accepted for publication. Below are some comments for improving the manuscript:

1.     Figure caption should be able to stand alone for describing the figure without referring to the main text. The sub-figures in Fig 1 should be labeled with (a), (b), (c) & (d), and each sub-figure should be described in the figure caption. Same issue occurs in Figs 2, 3, 4, & 9.

Moreover, there are wrong captions mistakenly put: Figs 11, 12, 19 & 20. Fig 10 is for target magnet, not master one.

This point must be corrected since the Production Editor will still ask the authors to correct and improve prior to publication.

2.     What does the circled dot in Figure 3 mean?

3.     Typo in Fig 8 & 10: Hall element, not Hole element. Moreover, Lines 151 & 243 say the active area of the Hall element is 50 um x 50 um, but this two Figures say 0.5 x 1.2 (without unit) – please confirm and correct.

4.     Some unsuitable editing issue:

Line 232: in Table III, should be Table 3

Line 235 & 237: in Section III.B, should be 3.1.2

Line 243: 50-um2?

Line 334: in Table I, should be Table 1

5.     Lines 157-159: the current description is too complex in gramma, and is better divided into two sentences. For example: The laser scanning speed which results in the maximum surface flux density was applied to magnetize the master magnet. The peak-to-peak (p-p) values of the pole pairs having maximum surface flux density were measured near the center of each region in the ranges of 0.9–1.5 mm and 3.5–4.1 mm.

In this way, following the descriptions, the authors should/could explain the reason why the evaluation adopted only the data in the blue regions, not the entire polarities, as shown in Fig 5.

6.     Please specify the grade class or type symbol of the SUS.

7.     Lines 254-255: As shown in Fig 11, …. as the heating temperature increased and …. But such a description is not shown in Fig 11.

8.     Explain the reason why the thermal stress analysis is conducted with the SmCo/PI/NdFeB sandwich structure, and why 3 mm in length is analyzed.

9.     Comparing the stripe and checkerboard patterns in Figs 15& 18, the polarities in target magnet are much more distinguishable and complete than those in master magnet. However, the values in Tables 5 & 6 (Line 340 should be Table 6) couldn’t reveal such a result. Since there is an offset issue as shown in Fig 17, RMS is not a good inspector for comparison. The authors have to fix such an issue to precisely present the results in a quantitative way.

10.Continued above, the ideal results in Fig 15 (master magnet) and Fig 18 (target magnet) are exactly the same, facing a dilemma of simulating ideal case. The authors must explain and discuss more differences (and/or possible differences) between these two “ideal cases” (what should be simulated/included but not yet done in the current study) so that readers can avoid the issue of misleading baseline when comparing the results.

11. Line 320: 4.3 should be 4.4

12. Line 330: where do the Br = 0.44 T and Hc = 356.8 kA/m come from?

13. Sentences that are difficult to read, especially for entry-level researcher:

Lines: 114-118

Lines: 258-259

Lines: 276-279

Lines: 310-312

Line: 363

Lines 21-25 in Abstract: better rephrase to specify the grades of master and target magnets

14. Grammatical error:

Line 60: can be is

Line 75: eliminates overcomes

Line 78: elaborate on

Line 172: which and was

Line 206: An SUS, not a SUS

Comments on the Quality of English Language

This manuscript must be edited by a native English speaker after revision, prior to re-submission.

Author Response

Thank you very much for taking the time to review this manuscript. Please see the attachment.

Reviewer 2 Report

Comments and Suggestions for Authors

In the manuscript, MPT approach is verified in transferring the pattern for the magnetic MEMS device. Overall, there are some merits in the manuscript. However, the following issues should be addressed:

1. There are a few writing issues in the manuscript. To name only a few in the following:

(1)The desired alternating magnetic pattern can be is formed by repeating this method. 

(2)This method eliminates overcomes the limitations of magnetic patterns by using fine and complex magnetization patterns fabricated using the LAH magnetization method as a substitute for SmCo magnet arrays in the UHM process.

(3) Fig. 9 shows the MPT test setup and process. The experiment was performed at room temperature at room temperature (22 °C) and the atmospheric pressure. 

2. In Fig.20, for the hysteresis loops of the sample, the sample's orientation in the magnetic field should be clarrified.

3.The influence of the temperature on the pattern transfer should be discussed in detail. For example, as shown in Fig. 17, under heating at 160 °C, the p-p value of the surface flux density was maximized, and the offset value was relatively close to zero. Please clarify the concrete reason.

Comments on the Quality of English Language

English is fine except for a couple of grammar mistakes and typos.

Author Response

(The authors gave the same response as above.)

Round 2

Reviewer 1 Report

Comments and Suggestions for Authors

Thanks for the revisions. Great work.

Reviewer 2 Report

Comments and Suggestions for Authors After reading the revision and the reply, I am glad to see that all my concerns have been addressed. Comments on the Quality of English Language

English is fine except for a couple of grammar mistakes and typos.